



# Towards geologically reasonable lithological classification from integrated geophysical inverse modelling: methodology and application case

Jérémie Giraud[1], Mark Lindsay[1], Mark Jessell[1], and Vitaliy Ogarko[2].

[1] Centre for Exploration Targeting (School of Earth Sciences), University of Western Australia, 35 Stirling Highway, 6009 Crawley, Australia.
[2] The International Centre for Radio Astronomy Research, The University of Western Australia, 7 Fairway, 6009 Crawley, Australia.

**Correspondence:** Jérémie Giraud (jeremie.giraud@uwa.edu.au)

**Abstract**. We propose a methodology for the recovery of lithologies from geological and geophysical modelling results and apply it field data. Our technique relies on classification using self-organizing maps (SOM) paired with geoscientific consistency checks and uncertainty analysis. In the procedure we develop, the SOM is trained using prior geological information in the form of geological uncertainty, the expected spatial distribution of petrophysical properties, and constrained geophysical inversion results. We ensure local geological plausibility in

the lithological model recovered from classification by enforcing basic topological rules through a process called 'post-regularisation'. This prevents the three-dimensional lithological model from violating elementary geological principles while maintaining geophysical consistency. Interpretation of the resulting lithologies is complemented by the estimation of the uncertainty associated to the different nodes of the trained SOM. The application case we investigate uses data and models from the Yerrida Basin (Western Australia). Our results generally corroborate

previous models of the region but they also suggest that the structural setting in some areas need to be updated. In particular, our results suggest the thinning of one of the greenstone belts in the area may be related to a deep structure not sampled by surface geological measurements and which was absent in previous geological models.

## 1.     Introduction

The idea of geoscientific integration is not new and has been advocated since the inception of quantitative

geoscientific studies involving geophysics and during the advances of geophysics as a discipline (see for instance Wegener, 1915, Nettleton, 1949, Towles, 1952, Jupp and Vozoff, 1975, Lines et al., 1988, Li and Oldenburg, 2000, ). In the natural resource sector, the exploitation of the inherent duality of geology and geophysics has been recognized as one of the pre-requisites to exploration success as early as the 1940's during the early years of the Society of Exploration Geophysicists (Eckhardt, 1940, Green, 1948). Numerous authors have since tackled the

issue of integrating petrophysical and geological information to model geophysical quantities through inversion, with an increasing trend in the past 15 years or so (see for instance references reviewed in Lelièvre and Farquharson, 2016, Meju and Gallardo, 2016, Moorkamp et al., 2016, Giraud et al., 2017). In contrast, the recovery of geological quantities from geophysical inverse modelling has seen much less effort. Recent studies have started to rectify this by proposing the idea of lithological differentiation of inverse modelling results

(Paasche et al. 2010, Sun and Li 2015, Paasche and Tronicke, 2007), which consist in the identification of



lithologies from inversion results. While lithological differentiation is expected to hold much potential in mineral exploration it still remains underexplored (Li et al., 2019),.

In oil and gas exploration scenarios, seismic facies analyses and classification using techniques developed for what is commonly called *machine learning* (e.g., neural networks, support vector machine algorithms, etc.) have

become popular in recent years (Zhao et al., 2015, Chopra and Marfurt, 2018, Wrona et al., 2018, Zhang et al., 2018). This was driven, on the one hand, by the need for quantitative interpretation methods in the geosciences, and by the 'renaissance' phase machine learning went through after 2006 (Chap. 5, Goodfellow et al., 2016) on the other hand. Once recovered, spatial facies distribution can be used for geological interpretation and downstream decision-making. Nevertheless, like all modelling results, the identification of facies or lithologies

using machine learning relies upon statistical models and is affected by ambiguity and uncertainty. One reason for this is that validation datasets are usually treated as the ground 'truth' while they are fraught with uncertainty. For instance, the interpretation of borehole data or outcrops with their uncertainty can lead to significantly different models honouring geological measurements equally well (Wellmann et al.,2010,de la Varga et al.,2018,Pakyuz-Charrier et al., 2018a, b, c,, ).

It is also evident that the modelling of lithology (or facies) is sensitive to a broad range of parameters and that their characterization relies on properties that are the results of complex, non-linear physical processes. In this context, one possible solution is to use neural networks for lithological classification as they are "universal approximators" (van der Baan and Jutten, 2000 ). As a consequence, however, lithological classification is affected by uncertainty from the data used to train and validate the algorithm. Such uncertainty is difficult to quantify and

is rarely estimated or even considered.

To date, whether it be in oil and gas or mining exploration, uncertainty in recovered rock types is a research avenue, which, to the best of our knowledge, only a few authors have addressed. (Sun and Li, 2019) assess uncertainty by varying the number of clusters in their lithological differentiation scheme, and (Bauer et al., 2003) classify lithologies and estimate the resolution of their results using synthetic data. As a result of the lack of

comprehensive uncertainty analyses, practitioners often lack quantitative, robust uncertainty modelling necessary to informed interpretation or risk evaluation (Jessell et al., 2018). In addition, apart from (Zhao et al., 2017) who account for seismic data-driven stratigraphy in their seismic facies classification, established workflows relying on neural networks to identify facieses or rock types in three-dimensions give little to no consideration geological information and rules for their classification.

To mitigate this, and to complement existing methodologies, we propose a solution that partially addresses the issues and shortcomings highlighted above. We introduce a general post-processing (i.e., post-inversion) workflow for the recovery of lithologies from geoscientific modelling results and the estimation of the related uncertainty. For this purpose, we complement existing classification techniques by ensuring the geological and geophysical consistency of, and estimating the confidence in, the recovered lithologies. Using an artificial neural

network trained in a fully controlled environment with attributes characterizing the geophysical inverse model, we perform lithological classification applying plausibility filters relying on geological principles which we refer to as 'geological post-regularisation'. The application of geological post-regularisation is to reduce the non-geological character of models obtained through classification. After classification, we calculate the probability



of occurrence of the different lithologies and estimate the related uncertainty for each model-cell discretizing the studied area.

The methodology we propose can serve multiple objectives. Our first objective is to introduce a methodology that is made efficient by leveraging existing geoscientific inputs and prior information, and cost-effective by imposing
requirements that do not exceed the computational power available on a personal computer. Secondly, our aim is to complement inverse modelling workflows by providing a general, automated method to derive a probabilistic lithological interpretation of inverse modelling results. Thirdly, we propose a real-world application based on a case study in the Yerrida Basin (Western Australia) where we build upon recent work by (Giraud et al., 2019a) and (Lindsay et al., 2018) who performed the geophysical inversion and geological modelling of data collected in
the area, respectively.

In this method, lithologies are identified though a classification technique relying on a simple artificial neural network. We chose 'self-organizing maps' (Kohonen, 1982), a well-established algorithm that has been successfully applied to seismic facies classification and geological mapping purposes (Chang et al., 2002, Klose, 2006, Köhler et al., 2010, Bauer et al., 2012, Carneiro et al., 2012, Du et al., 2015, Roden et al., 2015). We first
train and test the self-organizing maps (SOM) using data extracted from a semi-synthetic dataset (i.e., a geophysical inversion feasibility study based on geological and petrophysical field data) assumed to represent the geophysical characteristics of the studied area. We utilize this controlled environment to estimate the accuracy and uncertainty of our predictions for each class identified in the studied volume. We then use the trained network to perform classification using field data inversion and modelling only. We obtain, for each model cell, a suite of
probabilities equal to the relative frequency probability of occurrence of each lithology observed in the area. In both cases, geological post-regularisation is applied before the calculation of uncertainty metrics to ensure the geological consistency of the results.

The methodology is summarized in Figure 1 below.

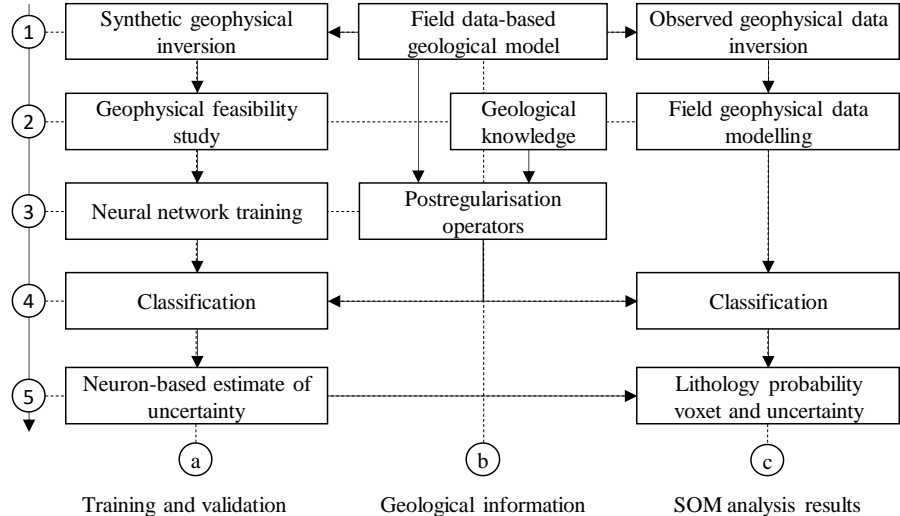

**Figure 1.** Schematic summary of proposed methodology.





## 2.    Methodology for geoscientific modelling and classification

In this section, we first introduce essential information about the geophysical and geological modelling used as pre-requisite to this study. We then introduce the utilisation of SOM and the tools we developed in sufficient detail to allow the reproducibility of the procedure.

### 2.1    Integrated geophysical modelling

### 2.1.1    Geophysical inversion scheme

Inverse geophysical modelling was performed using the least-square inversion platform Tomofast-x. This inversion platform enables the use of a series of constraints as detailed in Martin et al. (2018), Giraud et al. (2019a), (2019b). Constraints are enforced through a minimum-structure gradient regularisation approach which weight

varies locally accordingly with geological uncertainty (Giraud et al., 2019a). The cost function to minimize is given as:

$$\theta(d, m) = \left\| W_d\big(d - g(m)\big) \right\|_2^2 + \alpha_m \left\| W_m\big(m - m_p\big) \right\|_2^2 + \alpha_H \| W_H \nabla m \|_2^2, \tag{1}$$

where $d$ represents the measurements and $m$ is the model; $g$ is the forward operator calculating the data $m$ produces; $m_p$ is the prior model. In this contribution, $W_d$ is a diagonal matrix where each element is equal to the inverse of the sum-of-squares of the geophysical measurements; $W_m$ and $W_H$ are diagonal covariance matrices;

$\alpha_m$ and $\alpha_H$ are weights controlling the relative importance of the different terms in the equation; $\nabla$ is the spatial gradient operator. The last term of the equation (1) constraints the structural features of the inverted model. The values in diagonal matrix $W_H$ are determined from prior information. In this work, $W_H$ is obtained from geological modelling results and is a proxy for geological uncertainty.

The matrix $W_H$ is calculated following (Giraud et al., 2019a), who use the probabilistic geological modelling

approach described in Pakyuz-Charrier et al. (2018c, 2018b, 2019). In the case of gravity inversion as presented here, the complete Bouguer anomaly of density contrast model $m$ is calculated as the product of the Jacobian matrix $G$ with model $m$, i.e., we have $g(m) = Gm$.

### 2.1.2    Geological uncertainty

Geological uncertainty is estimated from probabilistic geological modelling. During this process, a series of

geological models is generated using the Monte-Carlo Uncertainty Estimator (MCUE) of Pakyuz-Charrier et al. (2018a, 2018b, 2018c, 2019). MCUE relies on the perturbation of orientation measurements (interfaces and foliations) defining structures of a reference geological model accordingly with their uncertainty. From this series of models, geological uncertainty can be estimated through calculation of Shannon's entropy (Shannon, 1948) for the simulated geological models (Wellmann and Regenauer-Lieb, 2012). This metric, which can be used as a

proxy for geological uncertainty, indicates how well geological information constraints the model locally. It can be used to constrain inversion in a structural sense when integrated in $W_H$ as per equation (1) (Giraud et al., 2019a).

More detailed information about the usage of MCUE results in geophysical inversion can be found in Giraud et al. (2017, 2018, 2019a, 2019b).





### 2.2 Classification using SOM and uncertainty analysis

#### 2.2.1 Fundamentals of self-organizing maps

The artificial neural network called 'self-organizing maps' (Kohonen, 1982a, 1982b) (SOM) relies on soft competitive, non-supervised learning. The relative simplicity and the efficiency of the SOM algorithm has made

it a popular tool for classification, data imputation, visualization and dimensionality reduction (Vesanto and Alhoniemi, 2000, Kalteh et al., 2008, Miljkovic, 2017, Klose, 2006, Kohonen, 1998, 2013, Roden et al., 2015, and Martin and Obermayer, 2009). In essence, it consists in the projection of the SOM's (usually two-dimensional) latent space onto a manifold of superior dimension (i.e., our dataset). This map is made of a predefined number of interconnected neurons (also referred to as 'nodes' or 'units') that have a fixed network configuration.

Projection occurs during the training phase, where the locations of the neurons in the manifold are iteratively adjusted so that they approximate it optimally.

#### 2.2.2 Training and validation process

##### 2.2.2.1 SOM specifications and training procedure

In this study, we follow common practice by training two-dimensional (2D) maps using a hexagonal lattice

topology and applying a Gaussian-shaped neighbourhood function. We chose to use a 2D map for the sake of simplicity after our testing revealed that other configurations did not improve results significantly. The hexagonal lattice topology seemed to provide better results than square lattice topology using the dataset we present here.

In the approach we follow, the optimum number of neurons (or units) is determined using the elbow curve of the mean quantization error Q (see equation 2 below in the next subsection) of the trained SOM. Note that this

approach follows the same principle as the well-known L-curve principle (Hansen and O'Leary, 1993, Hansen and Johnston, 2001, Santos and Bassrei, 2007) for the determination of optimum weights in least-square geophysical inversion. Here, we train the SOM using functions from the SOM Matlab toolbox implemented by (Vatanen et al., 2015).

##### 2.2.2.2 Training and testing datasets

Ideally, the SOM should be trained in a controlled environment where all the variables used are perfectly known. This motivates the utilisation of synthetic geophysical data. In this work, we calculate such data from a geological structural framework derived from real-world field geological and petrophysical field measurements data in the same fashion as for a geophysical feasibility study (Figure 1, lines 1 and 2). The training and tests datasets are comprised of the following variables:

1. Starting model for inversion $m_{start}$;

    2. Inverted model $m_{inv}$;

    3. Geological uncertainty $W_H$;

    4. Spatial gradient in the inverted model $\|\nabla m_{inv}\|_2$;

    5. Most likely lithology obtained from geological modelling.

Each datum is a vector $x \in \mathbb{R}^{n_v}$, with a number of variables $n_v = 5$, where $u(5)$ is the lithology assigned to this unit. This choice of training variables was motivated by the necessity to account for the available information in





terms of geological modelling and measurements, uncertainty and structural setting, while using interpretable datasets. The starting model for inversion encapsulates the pre-inversion state of knowledge. The inverted model comprises the update of this model using information extracted from geophysical measurements and translated into a 3D model. The lithological model refers directly to the interpreted geological observations of the area. The

spatial gradients of the inverted models provide structural information about the location of the geological units that can be recovered by interpretation from the inverted density contrast model.

During training, we examine SOM quality using quantization error $Q$ and lithology prediction accuracy. Quantization error "measures the average distance between each data vector and its best matching unit [BMU]" (Uriarte and Martín, 2005), thereby indicating how well the different BMUs approximate the dataset. It can be

interpreted as analogous to a misfit between calculated and observed data. The mean quantization error $Q$ of the SOM is expressed as follows for $n$ data vectors $x_i$:

$$Q(\boldsymbol{x}, \boldsymbol{BMU}) = \frac{1}{n} \sum_{i=1}^{n} \|BMU_i - x_i\|_2^2,$$ (2)

where $BMU_i$ is the BMU of $x_i$. In the application of SOM to field data, the trained map is used for the classification of inversion results where input 1 through 4 listed above are obtained from previous modelling and lithologies are

the quantity sought for.

### 2.2.3    Geological post-regularisation

#### 2.2.3.1    Motivations

Geological rules have the potential to provide an important constraint on the classification of lithologies recovered from inversion. Such rules, like adjacency (Egenhofer and Herring 1990) define which rock bodies can be in

contact with each other, and which are cannot. These rules are typically expressed in geological terms as stratigraphy, where the relative age and event classification of geological units are stated. For example, a sedimentary depositional event of five separate units may define a simple sub-horizontal layer cake configuration where the oldest unit is never adjacent (or in contact) with the youngest unit. A magmatic event that follows may result in a vertical dyke that intrudes all sedimentary layers adjacent to all other rock units. Using geological rules

as a constraint relies on finding those that are restrictive (such as the youngest unit never being in contact with the oldest), rather than permissive (such as the intruding dyke). Thiele et al. (2016), Pellerin et al. (2017), Anquez et al. (2019) show how these can constrain parametric geological modelling. It is therefore important to honour geological rules if known, and include them in classification schemes such as those to ensure that geological plausibility is not compromised in pursuit of an otherwise petrophysically and geophysically consistent model.

The process of postregularisation, which consists in the application of spatial-contextual filters to the classification results to eliminate geologically unrealistic features, has been shown to increase prediction accuracy in surface (2D) geological mapping (Tarabalka et al., 2009, Stavrakoudis et al., 2014, Cracknell and Reading, 2015). s

#### 2.2.3.2    Implementation

The post-regularisation (PR) scheme we develop for the recovery of lithologies in 3D relies on two hypotheses.

Firstly, we assume that the presence of isolated lithologies contradicts the geological principle of continuity.





Although such post-regularisation has been used mostly in 2D or shallow 3D, there is no theoretical obstacle to the extension of this methodology to the purely 3D classification case we present here. Secondly, we introduce the utilisation of adjacency relationships between the different lithologies in post-regularisation to ensure that base topological rules are respected across the entirety of the three-dimensional volume. This is particularly important for structural geological interpretation (Freeman et al., 2010, Godefroy et al., 2019). In this paper, we extend existing post-regularisation approaches (ie,, Tarabalka et al., 2009, Stavrakoudis et al., 2014, Cracknell and Reading, 2015) by integrating geological information in the classification analysis in the form of topological relationships (see Egenhofer and Herring 1990, Zlatanova, 2000, and Thiele et al., 2016, for the different topologies) defined by geological principles.

The general formulation of post-regularisation is as follows, for a given model-cell:

$$BMU := BMU_k, \text{where } k = \arg\min(\|u_k - BMU\| \mid conditions), \tag{3}$$

where arg min returns the argument $k$ satisfying the conditions it precedes. Here, we set:

$$conditions = \{\exists\, c \subset U \mid c = BMU_k (5) \cap \mathbf{1}^T M^k \circ M^f \mathbf{1} = \mathbf{0}_{n_l,n_l} \cap stratigraphy\}, \tag{4}$$

where $\mathbf{1}$ is $n_l \times 1$ column vector of ones, and $\mathbf{0}_{n_l,n_l}$ is the $n_l \times n_l$ matrix of zeros; $M^f$ and $M^k$ are adjacency matrices, and $M^k \circ M^f$ is the Hadamard product of $M^k$ and $M^f$, such that $(M^k \circ M^f)_{ij} = (M^k)_{ij}(M^f)_{ij}$, and $\mathbf{0}_{n_l,n_l}$ is the zero matrix of dimension $n_l \times n_l$; $M^f$ encapsulates geological knowledge and principles about contacts between lithologies; $M^k$ contains the adjacency relationships between the considered cell and its neighbourhood $U$. The derivation of $M^f$ and $M^k$ is detailed below.

The first part of the condition in equation (4) is enforced using morphological closing where isolated lithologies are replaced by the most prevalent one in their neighbourhood (see Benavent et al., 2012, Ackora-Prah et al., 2015). In such cases, the BMU is updated as follows. Isolated cells (in terms of their lithology) are identified through examination of the 26-cell 3D cubic Moore neighbourhood $U$ of every model-cell. A cell is considered isolated if, and only if, at least 25 cells of $U$ have a lithology that differs from it. Once such a cell is identified, its BMU is updated using the closest neuron ensuring continuity between adjacent cells in this neighbourhood $U$, where the lithology to be assigned is determined by a majority vote in $U$ subject to adjacency conditions. These conditions are determined by geological knowledge (as explained below). This process is repeated for all locations until the lithological model stops changing. The general principle of post-regularisation is illustrated in Figure 2.

We point out that in contrast to (Tarabalka et al., 2009) and (Stavrakoudis et al., 2014) who use the first and second Chamfer neighbourhoods in 2D around the considered model cell, we do not follow the same approach in 3D. Our implementation of the extension of their approach to 3D showed that, in our application case study, the adjustments of the recovered lithological model it imposes are detrimental to the consistency of the classification with geophysical measurements. That is, the perturbation of the corresponding geophysical response of the model it generates exceeds noise level and compromises the geophysical validity of the recovered model (see geophysical validation subsection below for more details). The same remark applies to the utilisation of a mode filter with a 3×3×3 kernel.





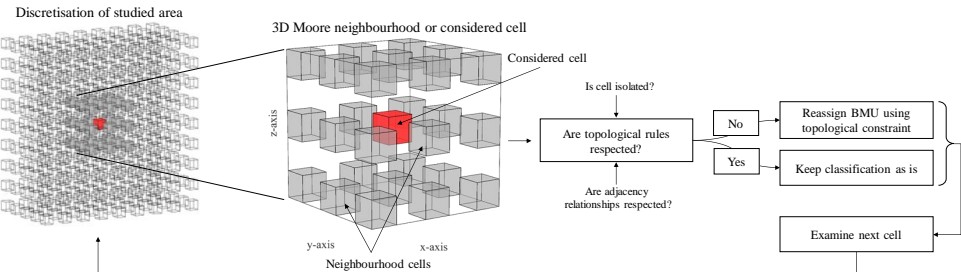

**Figure 2.** Summary of topological filtering used during post-regularisation.

The conditions relating to adjacency relationships forces the model to honour adjacency relationships extracted from surface geology (Burns, 1988, Thiele et al., 2016) in the recovered lithological model.

We determine lithological topology by identifying the contacts between adjacent model-cells and represent the topological signature of lithological models using the adjacency representation of (Godsil and Royle, 2001). Let the adjacency matrix $M$ of a given cell be defined as:

$$M_{i>j,j} = n_{ij}, M_{i \leq j,j} = 0,  \tag{5}$$

where $n_{ij}$ is the number of contacts between lithologies $i$ and $j$. Similarly, geological laws and knowledge allow the derivation of a matrix $M^f$ defined as follows:

$$\begin{cases} M^f_{i>j,j} = 1 \; if \; contact \; between \; i \; and \; j \; contradicts \; geology. \\ \qquad M^f_{i>j,j} = 0 \; otherwise \end{cases}  \tag{6}$$

From there, it is straightforward to identify occurrences of forbidden contacts by calculating $M_k \circ M^f$. Therefore, $\mathbf{1}^T M_k \circ M^f \mathbf{1} = 0$ (equation 4) indicates that no contact violating the condition imposed by $M^f$ is observed. The last condition in equation (4) can be used to prevent the local stratigraphy (in the Moore neighbourhood $U$ of the considered cell) from violating geological rules such as, for instance, "lithology B must always be between lithologies A and C in a conformable sequence".

After identification of configurations forbidden by the conditions set in equation (4), its BMU is updated using the closest neuron honouring the set of conditions (Figure 1, box 4-a).

The next stage of the methodology we introduce is the calculation of the apportionment of each neuron in terms of the lithologies of the testing data vectors (from the synthetic survey) they predict (Figure 1, box 5-a). For instance, in a two-lithology scenario, a given node may be predict lithology A using the validation dataset with
80% accuracy with index B with 20% accuracy. This process is described below.

### 2.2.4    Estimation the confidence in recovered lithologies

The prediction accuracy $\tau_i$ of lithology $i \in [1, \dots, n_l]$ (with $n_l$ the total number of lithologies) is the ratio of correct predictions to the total number of predictions. It is obtained from the 'matching matrix' of the recovered lithologies $M^c$. We remind that the 'matching matrix' (or 'confusion matrix' in supervised learning) is a matricial
representation of the number of occurrences of true/false positives/negatives.





We use the prediction accuracy $\tau_i$ as a metric measuring the capability of the nodes of the trained SOM to recover lithologies. Let $\tau_i$ be expressed as:

$$\tau_i = \frac{M_{ii}^c}{M_{ii}^c + \sum_{\substack{j \neq i \\ j=1}}^{n_l} M_{ij}^c},$$

(7)

which is a particular case of the overall accuracy $\tau$:

$$\tau = \frac{\sum_{i=1}^{n_l} M_{ii}^c}{\sum_{i=1}^{n_l} \sum_{j=1}^{n_l} M_{ij}^c},$$

(8)

From equation (7), it appears that $\tau_i$ is equivalent to the relative apparition frequency of the $i^{\text{th}}$ lithology over the entire SOM. When considering a specific node $j$, it becomes the relative apparition frequency of the lithology $i$ for cells classified as having the $j^{\text{th}}$ node as their BMU, noted $\tau_{ij}$. These percentages, which we address as probabilities, constitute apparition frequencies. Moreover, the nodes of the trained SOM each approximate a subset of the dataset that vary in size. This issue is addressed below in the next subsection.

We complement the estimation of the probability of recovering the modelled lithologies by calculating uncertainty on these probabilities (Figure 1, box 5c). We calculate the Wald-type confidence interval (Wilson, 1927, DasGupta et al., 2001) of the recovered apparition frequencies, which we address here as probabilities. Details about the calculation of the confidence interval are given in Appendix A.

### 2.2.5    Geophysical consistency

The consistency of the classification performed using SOM after application of post-regularisation with field geophysical measurements might be altered by both the classification itself and by post-regularisation. It is therefore necessary to ensure that the approximation of the dataset by the units of SOM is consistent with geophysical measurements. To this end, we verify that the geophysical response of the density contrast model $\boldsymbol{m_{SOM}}$ corresponding to the BMUs of each cell in the studied area fits the field measurement $\boldsymbol{d}$ within a certain tolerance assumed to approximate noise level. Consequently, we ensure that the difference between $\boldsymbol{m_{inv}}$ and $\boldsymbol{m_{SOM}}$, $\Delta\boldsymbol{m} = \boldsymbol{m_{inv}} - \boldsymbol{m_{SOM}}$, satisfies the following condition:

$$\left\| \boldsymbol{W_d} \left( \left( \boldsymbol{d} - \boldsymbol{g}(\boldsymbol{m_{inv}}) \right) - \left( \boldsymbol{d} - \boldsymbol{g}(\boldsymbol{m_{SOM}}) \right) \right) \right\|_2^2 = \| \boldsymbol{W_d}\boldsymbol{G}\Delta\boldsymbol{m} \|_2^2 \leq tol,$$

(9)

where $\boldsymbol{tol}$ is the threshold depending on noise levels in the data above which $\boldsymbol{m_{inv}}$ and $\boldsymbol{m_{SOM}}$ are not considered geophysically equivalent.

The implication of equation (9) is that the difference $\Delta\boldsymbol{m}$ belongs to the null-space of the inverse problem considered. The null-space is characteristic of geophysical inversion's non-uniqueness. It is defined as the ensemble of models that reproduce geophysical data with a comparable misfit. The models honouring equation (9) can therefore be considered equivalent from a geophysical data point of view (Muñoz and Rath, 2006, Chen et al., 2007, Deal and Nolet, 1996).





### 3. Application case: Yerrida basin

### 3.1 Survey setting

This subsection introduces and summarises the geological and geophysical context of this study. More details about the geology of the area and the initial field data geophysical modelling can be found in (Giraud et al., 2019a) and (Lindsay et al., 2018).

#### 3.1.1 Geological and geophysical setting

The Paleoproterozoic Yerrida Basin is located in the southern part of the Capricorn Orogen (WA), and covers approximately 150km N-S and 180 km E-W ((Pirajno and Adamides, 2000)) (Figure 3a). The structures of interest in this work are Archean greenstone belts (Figure 3) as they are prospective for Au and Ni and underlie the younger basin rocks. Basement to the Yerrida Basin is considered to be Archean granite-gneiss or greenstone rocks of the Yilgarn Craton. Lithospheric extension initiated the formation of the Yerrida Basin at c. 2200 Ma to 1990 Ma with deposition of the Windplain Group (Occhipinti et al., 2017, Pirajno and Adamides, 2000). The Goodin Inlier remains exposed in the central part of the basin and is in unconformable contact with the Windplain Group. A hiatus ensued, followed by deposition of the younger Mooloogool Group, which was then overlain in the east by the Tooloo Group of the Earaheedy Basin.

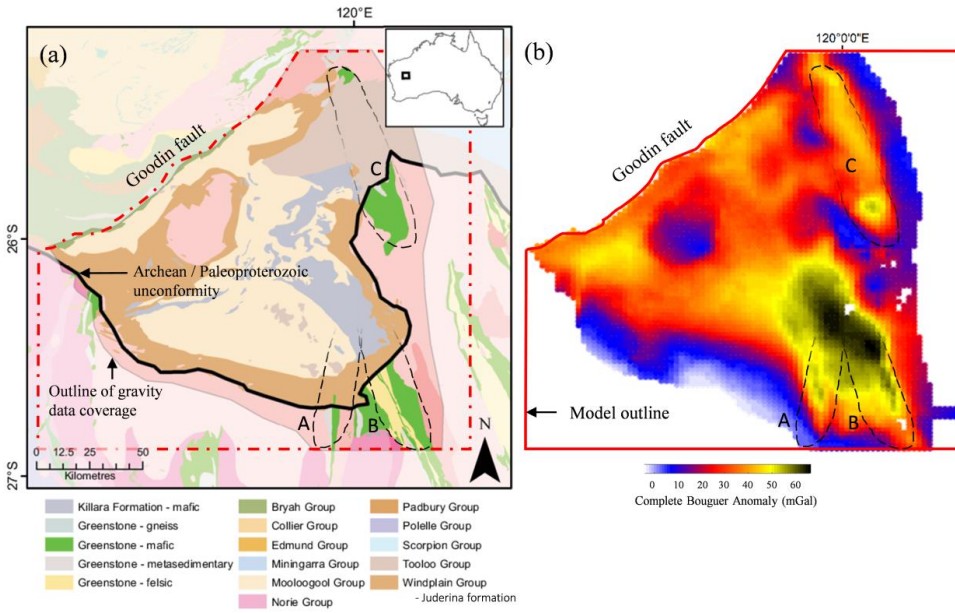

**Figure 3.** (a, to the left) Geological map of the area and (b, to the right) complete Bouguer anomaly (reproduced from Giraud et al., 2019a). The dashed lines delineate the possible sub-basin extent of the mafic greenstone belts. Capital letters 'A', 'B' and 'C' symbolize the possible outlines for the greenstone belts in the area.

The density contrast of the rock types observed in the area range between 0 and 330 kg/m³, making it appropriate for gravity modelling and inversion (Giraud et al., 2019a, Lindsay et al., 2018). While basin rocks exhibit some density contrast, the greenstones are conspicuous in gravity data with a density contrast expected to lie between 190 and 270 kg/m³ making them an attractive subject for gravity inversion. Field geological measurements

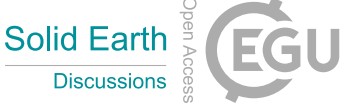

(orientation data in the form of interfaces and foliations) and petrophysical data were used to build the reference geological model. Airborne geophysical data, Landsat and Aster 8 satellite data were also used to support the interpretation of geological measurements.

The gravity anomaly dataset we consider (Figure 3b) is comprised of a total of 4882 measurement points. The
5   model is discretized into 100×100×42 cells of dimensions 2.335 km ×1.875 km ×1.0475 km down to approximately 44km depth. Weights and parameters used for the inversion of synthetic data follow the settings of (Giraud et al., 2019a) on field data.

### 3.1.2 Geological modelling and synthetic geophysical survey

This subsections introduces the semi-synthetic survey we performed for the training of SOM.

10   The volume of most probable lithology, $W_H$ and the starting model are shown in Figure 4. Volumes shown in Figure 4a, b, and c are used for the training and validation dataset for SOM training as explained in 2.2.2.2.

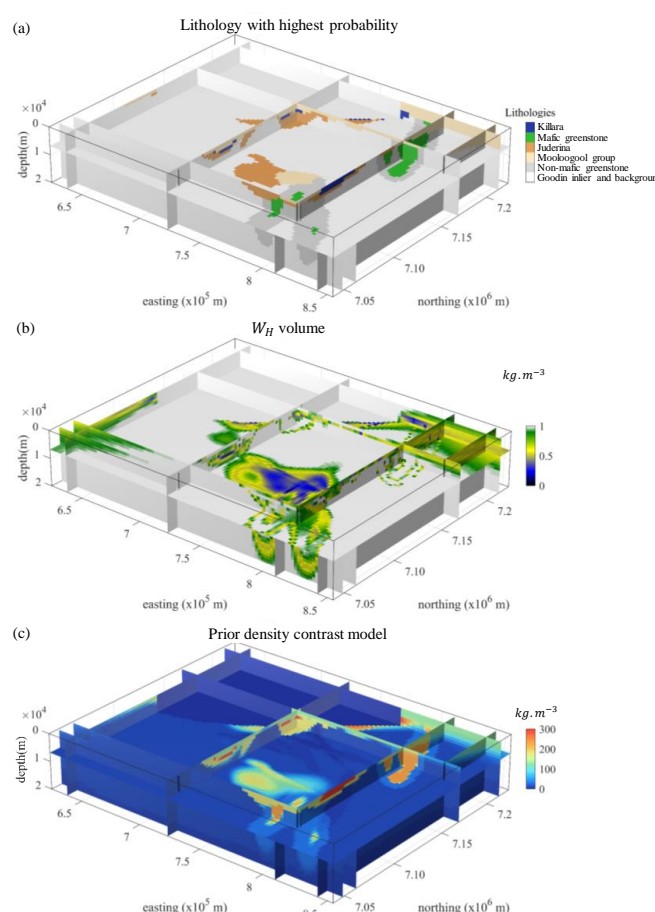

**Figure 4.** Prior geological modelling. Preferred lithology volume (a), geological uncertainty volume (b) and starting density contrast model (c). Modified from (Giraud et al., 2019a).



We use the modelling results shown in Figure 4 to calculate a synthetic geophysical dataset (Figure 5d). The model used to generate the synthetic geophysical measurements, the inverted model and its spatial gradients are shown in Figure 5a. The corresponding inverted model and its gradients are shown in Figure 5b and Figure 5c. Volumes shown in Figure 5b, and c are used for training and validation.

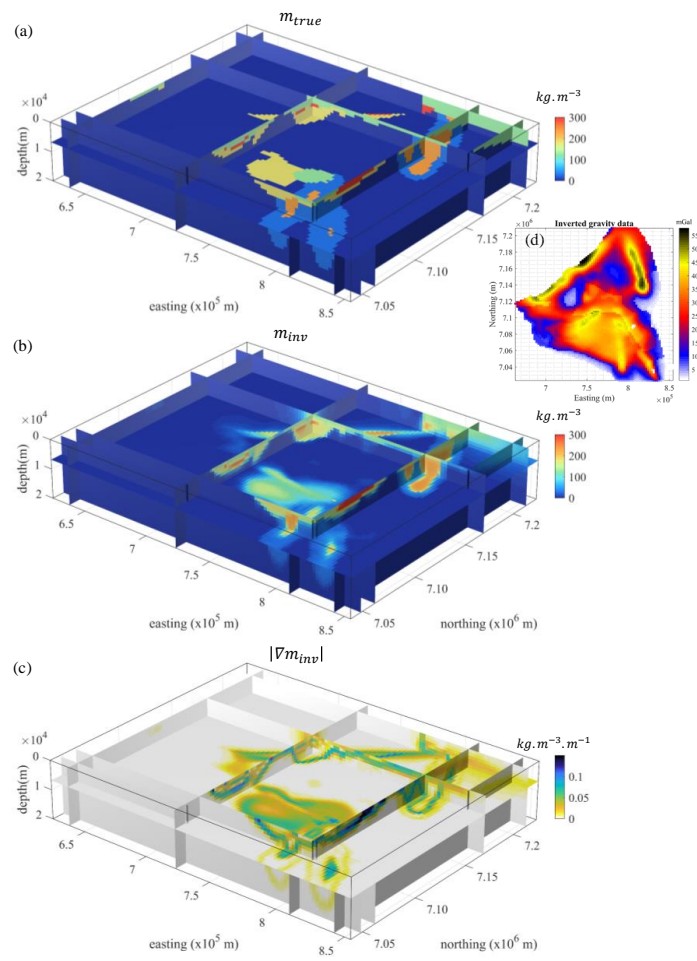

**Figure 5.** True density contrast model for calculation of synthetic geophysical data (a); inverted model obtained from integrated inversion of synthetic geophysical data (b), corresponding spatial gradient of density contrast (c) and synthetic geophysical data (d).

### 3.1.3    Field geophysical data inversion

The density contrast model obtained from inversion of field geophysical data and its gradients are shown in Figure 6a and b. Visual comparison of inverted models shown in Figure 6a and Figure 5b reveals that mesoscale structures are similar with the exception of large structures presenting low density contrasts at depth (darker shades of blue in Figure 6a). This is reflected in Figure 6b, which exhibits low gradient values in these areas.





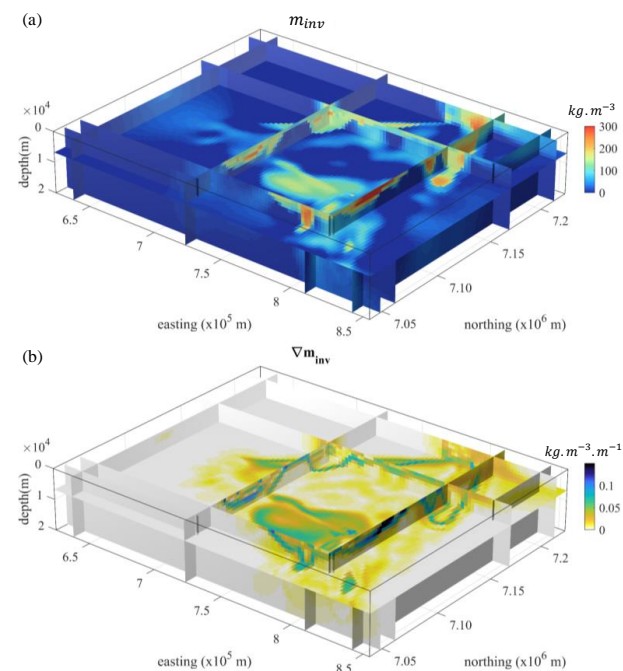

**Figure 6.** Inverted model obtained from integrated inversion of field geophysical data (a), corresponding spatial gradient of density contrast (b).

The classification of lithologies using SOM is performed applying the trained network to volumes shown in Figure
4b, c and Figure 6a, b. The next subsections describe the geological laws used for post-regularisation (subsection 3.1.4) and the classification process (subsection 3.2.2).

### 3.1.4  Geological rules for post-regularisation

As mentioned above, for clarity in this demonstration, only adjacency relationships are considered. This is also in part because in areas of sparse data, a full description of geological rules (fault relationships with fault and
stratigraphy) is often not known. Given the complexity of the Yerrida Basin and its magmatic and deformation history, several base geological rules can be derived to assess the plausibility of recovered lithological models. Using fundamental geological principles (such as uniformitarianism, superposition, Walther's Law, cross-cutting relationships and original horizontality), the two most likely restrictive adjacency rules are as follows. We assume that the mafic greenstone bodies cannot be in contact with the Killara Formation (in the Mooloogool Group) since
our field data suggests that the Killara Formation is a volcanic unit that is restricted to the Yerrida Basin and thus not in contact with the mafic greenstones. In addition, we assume that the mafic greenstones cannot be in contact with the Goodin Inlier and background (or basement) as the mafic greenstones are modelled to be enveloped by the felsic component of the greenstone.

The matrix $M^f$ defining the contacts forbidden by geology as described above is given in Figure 7.




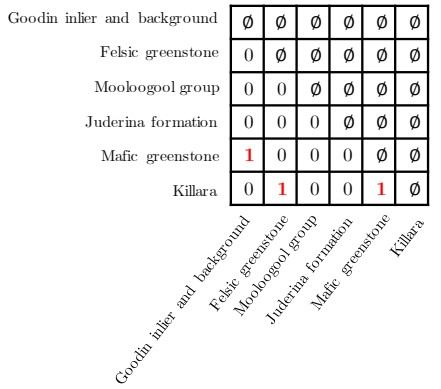

**Figure 7.** Matrix defining forbidding contact between lithologies in the Yerrida basin. 1 means that two units may be in contact with each other, 0 means that they may not and $\phi$ represents symmetric relationships or when the same unit is adjacent to itself (which geologically may occur across a fault, but cannot be resolved by the geophysical data available).

## 3.2 Geologically-constrained SOM classification and uncertainty analysis

### 3.2.1 Training the neural network

It this work, we use approximately 500 neurons (units) for the training of SOM. This number is inferred from the analysis of the elbow curve we calculated using the validation datasets (see Figure 8) and approximately matches the proposed value of $5\sqrt{N}$ (with $N$ the total number of observation) proposed by (Vesanto and Alhoniemi, 2000) and commonly used since (Shalaginov and Franke, 2015). The use of this value is corroborated by the lithology prediction accuracy (Figure 8) as approximating the point of diminishing returns, i.e., the number of nodes beyond which additional nodes are becoming nearly redundant.

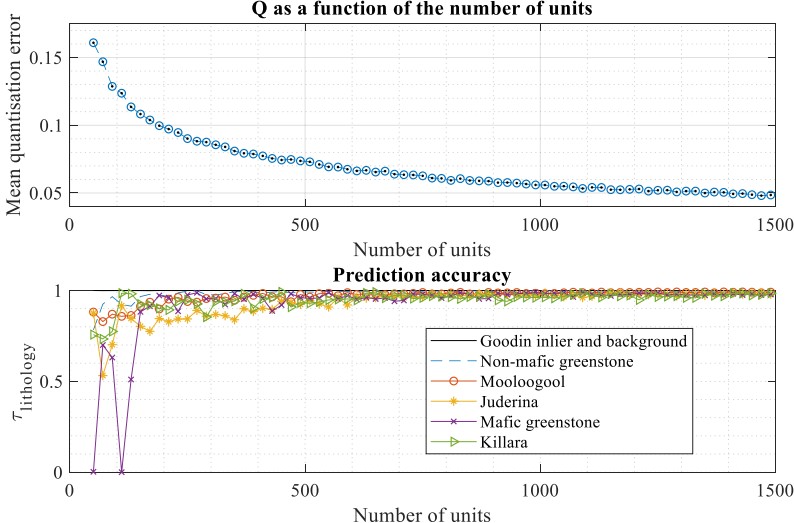

**Figure 8.** (a) Elbow curve of the quantization error for the determination of the optimum number of neurons (or units) in SOM and (b) Prediction accuracy for the different lithologies present in the training and validation datasets.





The trained map presents a mean quantization Q equal to 0.075, which indicates relatively good approximation of the datasets by the trained SOM. This is illustrated by Figure 8 where all lithologies are recovered with a prediction accuracy superior to 90%.

### 3.2.2    Classification and post-regularisation

5   After classification of the recovered model, we performed post-regularisation to remove geologically unrealistic features from the classified lithological volume. Figure 9 show the classification results before and after post-regularisation, along with the associated adjacency matrix.

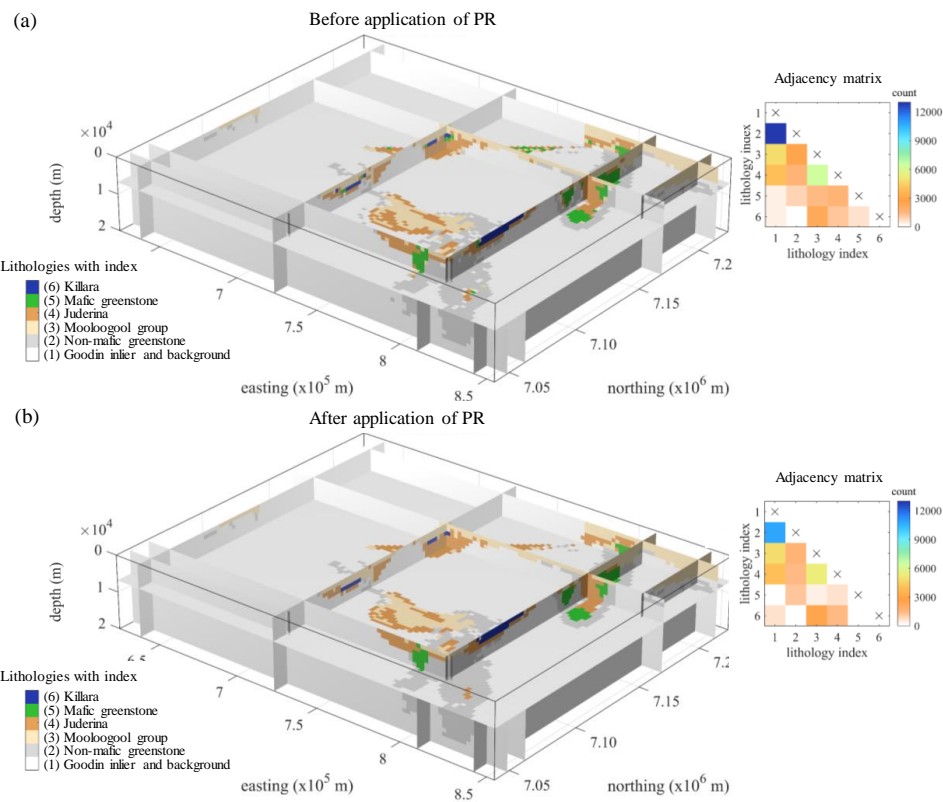

**Figure 9.** Recovered lithologies before post-regularisation (a) and after (b) next to the corresponding adjacency matrix (right hand side). The indices used to designate lithologies are indicated in the legend of the volume (left hand side).

As can be inferred from the adjacency matrices plotted in Figure 9, the number of contacts between units with indices 1 and 2, respectively, is reduced by the application of post-regularisation. One reason for this is the presence of a number of inclusions of lithologies 1 in 2 and vice versa. The number of contacts between lithologies 5 and 2 increased slightly due to post-regularisation because a large percentage of contacts between lithologies 5

15   and 1 have been re-assigned as contacts between 5 and 2. The elimination of contacts between 5 and 6 is also visible in Figure 9.





### 3.2.3 Estimated confidence in recovered lithologies

For each node of the SOM we calculate the prediction accuracy $\tau_i$ (equation 3) for the different lithologies observed in the area using the cross-validation dataset. After application of the trained SOM for the classification of inverse modelling results obtained from the inversion of field geophysical data, we obtain a 3D probability

5     volume for each lithology. Figure 10 shows the resulting probability volumes for the 6 lithologies present in the Yerrida basin.

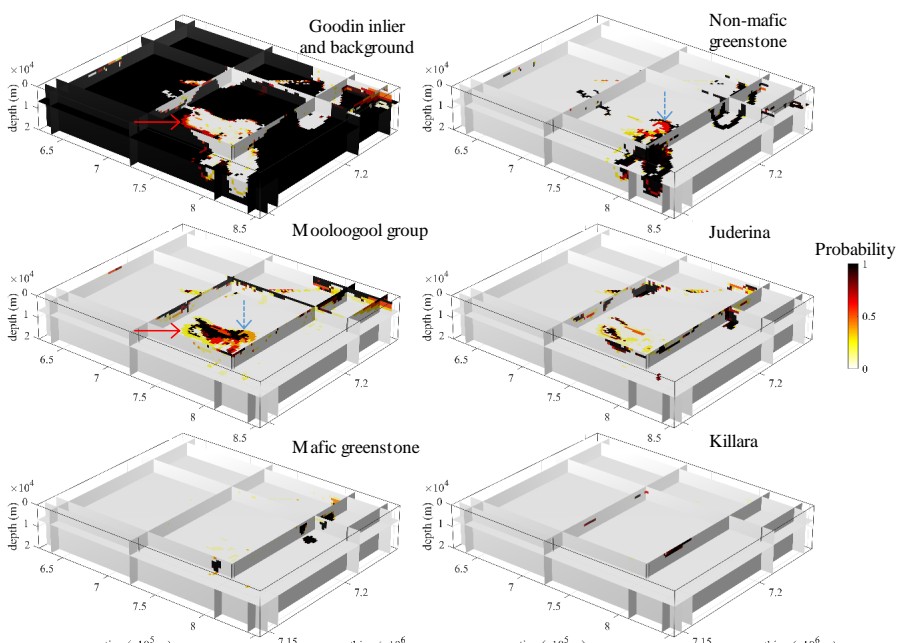

10   **Figure 10.** Probability volumes of recovered lithologies calculated as per equation (3). The arrows are drawn to support interpretation.

The uncertainty associated with the confidence level (equation 10) shown in Figure 10 is shown in Figure 11.

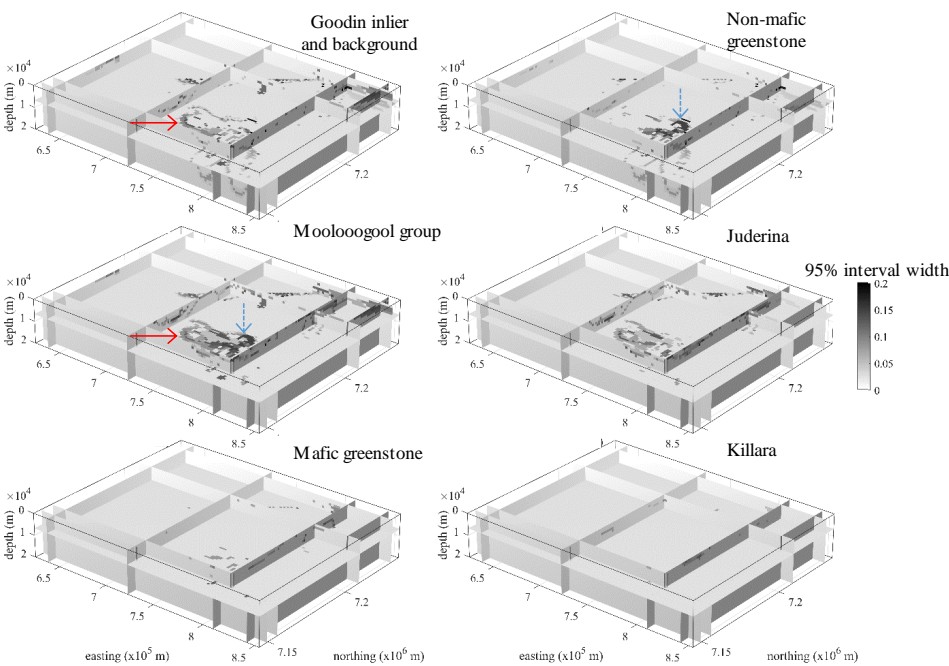

**Figure 11.** Value of 95% confidence interval of the probability of recovered lithologies, calculated as per equation (9).

A straightforward observation that can be made from Figure 11 is that uncertainty in recovered probabilities is very low for the majority of the model cells. It is non-negligible, however, in the central part of the model, where

variations in terms of density contrast and geological uncertainty are more pronounced. However, the values of the 95% confidence interval generally confirm interpretations that can be made directly from Figure 9 and Figure 10. Note that the zone of the model shown by the blue arrows in Figure 10 and Figure 11 exhibit probabilities between 0.3 and 0.6 on the former and 95% confidence intervals superior to 0.1. This suggests that these lithology probabilities in these zones are the least-well constrained. In contrast, the zone marked by the red arrow, which

also exhibit probabilities between 0.3 and 0.6, presents a narrow 95% confidence interval (<0.01), indicating that recovered lithology probabilities are less uncertain. Note that from Figure 10 and Figure 11 we can interpret the presence of mafic greenstone with confidence as this lithology is well constrained.

### 3.2.4    Geophysical null space validation and implications for geological interpretation

Applying equation (9), we obtain $\|W_d G \Delta m\|_2^2 = 6 \times 10^{-4}$, indicating that the model can be considered

geophysically equivalent overall. This indicates that we can consider the classified lithological model after application of post-regularisation as reflective of both geophysical and geological information. Focusing on the mafic greenstone belts of interest in the area (Figure 12), the classification results allow us to propose the following geological interpretations.

Figure 12 shows that the northern portion of the greenstone belts A and B recovered by geophysical inversion and

SOM classification is thinner in their Northern part than was proposed by the initial geological model. Likewise,



greenstone belt C seems to be much thinner near its centre than expected. Given the data density and lack of understanding we have about the depth of this greenstone belt, this observation is plausible. It confirms and refines considerably the crude, preliminary lithology differentiation of (Giraud et al., 2019a) that was based only on density contrast value. The cause of this thinning could be attributed to faulting, folding or to the topography of

the palaeoenvironment where the protoliths to the belt were formed, which are not captured directly by surface geology. The portion of the southern Merrie Greenstone Belt (mafic greenstone C) is shown to be thinner than expected, prompting a review of the structure of existing models. Plausible reasons range from the presence of structure that has not been identified from the initial interpretation of geophysical data. In addition, the potential presence of deep-penetrating faults or shear zones, as shown in Figure 12b by the arrows around greenstone C

hints a possible false assumption that the Merrie Greenstone Belt is one coherent geological body.

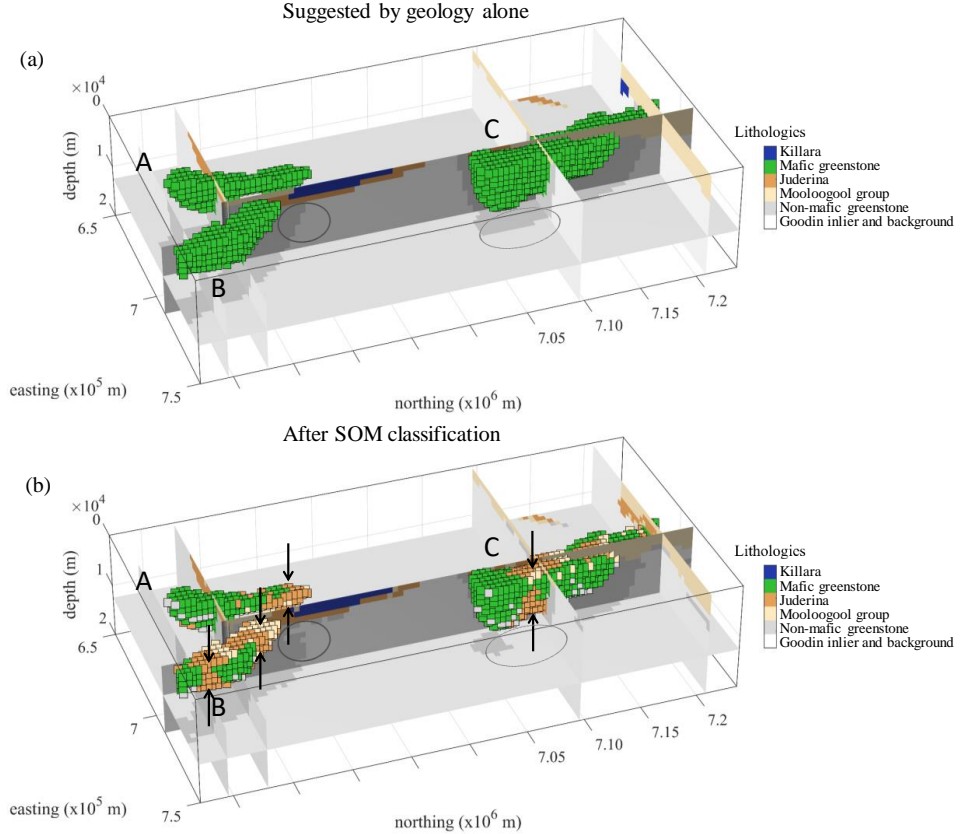

**Figure 12.** Mafic greenstone belts and their surroundings following geological modelling only (a) and after SOM classification (bottom). The cells shows in (a) and (b) have the same geographical location, and are coloured according to lithology. The vertical arrows show areas where mafic greenstone is thinner than suggested by geology only. The elliptical shapes shows

zones where expected non-mafic greenstone is replaced by the background lithology.





#### 4. Discussion

The application of the technique presented here is not restricted to usage of the particular geophysical or geological modelling schemes generating the modelling inputs to this study. The methodology we introduced is general and any different standalone geophysical and geological modelling schemes could also be used.

The work presented here relied on SOM, which can be seen as an extension of the k-means and c-means clustering algorithms used for lithological differentiation by (Paasche and Tronicke, 2007, Carter-McAuslan et al., 2015, Sun and Li, 2015, 2016b, Maag and Li, 2018, Ward et al., 2014, and Singh and Sharma, 2018), with which it shares a number of characteristics. We can therefore assume that our findings may hold true for these techniques.

We have shown that the utilisation of post-regularisation can be effective to increase geological realism in the
recovered lithological models while preserving the geophysical validity of the corresponding model. The geological principles we used to design our post-regularisation operator apply to lithological topology and focus on adjacency relationship between cells. Ideally, post-regularisation should also consider the surface area of contacts and their topology. This could be followed by, for instance, a 3D extension of the geological model-editing approach of (Anquez et al., 2019) to produce genuine geological models honouring age relationships,
stratigraphic principles, etc. Provided that models obtained in this fashion honour equation (9), this would ensure that while they are geophysically valid, they can be readily used for interpretation or by commercial or non-commercial geological modelling engines, reservoir simulations, etc. without further processing.

We also believe that post-regularisation can be successfully applied to other clustering techniques. In addition, the implementation of post-regularisation presented here can be readily applied to existing classification,
regardless of the classification algorithm used as it only adjusts the classification using spatial-contextual features in the classified model, and could assist the geological characterization of inversion results (Melo et al., 2017).

The example we have shown uses a covariance matrix $W_H$ (equation 1) that results from geological modelling. It is used as a proxy for the uncertainty about our knowledge in terms of structural geology. Such prior information could also be derived from techniques other than geological modelling such as, for instance, prior geophysical
modelling, be it using the same or different geophysical methods.

An important result produced here involves the identification of regions which do not adequately conform to the initial model parameters (Figure 12). While this issue remains unresolved, the capability of this method to identify problematic regions is useful to drive reinterpretation of data, consideration of additional models and, eventually, increased geological knowledge of the target.

Future work may include the generation of multiple lithological models using the trained SOM and the probability volume associated to it. By selecting models belonging to the null-space of the geophysical data (i.e., satisfying equation 9), we expect that this would allow the identification of a series of a few archetypes that would be representative of the various datasets used in the geoscientific modelling workflow.



## 5. Conclusions

We have introduced a post-inversion classification technique relying on SOM that enables the recovery of lithologies, the corresponding probability voxet and the associated uncertainty, thereby remediating to some of the limitations of deterministic inversion. The proposed technique utilizes a post-regularisation scheme enforcing elementary geological principles to the recovered lithological model while maintaining geophysical validity. We have applied this new methodology to the Yerrida basin (Western Australia) and shown how it improves the geological plausibility of the recovered model. This allowed us to confirm previous results and to bring new insights into possible reinterpretation of the geometry of prospective greenstone belts.

*Data availability.* The input and output of the synthetic survey are made available by (Giraud, 2019).

*Authors contribution.* Jérémie Giraud designed the methodology, adapted the SOM algorithm and performed all modelling except geological modelling. Jeremie is the main writer of the manuscript, which was redacted with support from the rest of the authors. Mark Lindsay performed geological modelling and interpretation of the recovered lithologies. Mark Jessell provided guidance and supervision while the project was being carried out. Vitaliy Ogarko assisted in the development of the parts of the methodology relating to geophysics and the writing of the paper on aspects relating to SOM.

*Competing interests.* The authors declare that they have no conflict of interest.

*Acknowledgements.* Vitaliy Ogarko acknowledges the Australian Research Council Centre of Excellence for All Sky Astrophysics in 3-D (ASTRO 3-D) for supporting some of his research efforts. Finally, the authors thank Evren Pakyuz-Charrier and Roland Martin for interesting discussions relating to topics covered in this paper.

*Financial support.* Mark W. Jessell was supported by a Western Australian fellowship. Mark D. Lindsay was supported by the Geological Survey of Western Australia and the Exploration Incentive Scheme and the Australia Research Council (DE190100431). Part of this work has been supported by an Australian Government International Postgraduate Research Scholarship. The authors acknowledge partial financial support from the MinEx Cooperative Research Centre. This research has been supported, in part, by LP170100985: Loop – Enabling Stochastic 3D Geological Modelling, funded by the Australian Research Council and supported by Monash University, University of Western Australia, Geoscience Australia, the Geological Surveys of Western Australia, Northern Territory, South Australia and New South Wales as well as the Research for Integrative Numerical Geology, the Université de Lorraine, RWTH Aachen, the Geological Survey of Canada, the British Geological Survey, the Bureau de Recherches Géologiques et Minières (French Geological Survey), and AuScope.



## 6.    Appendices

**Appendix A:** Uncertainty in estimated probabilities

We estimate the uncertainty attached to the association of an apparition frequency to a probability assuming that it is calculated as the average of a sufficiently large number of samples of a random variable. Under these

conditions, the Central Limit Theorem holds true (Sivia and Skilling, 2006, Brookes et al., 2007), meaning that this average can be described using a normal distribution. On this premise, we can calculate the Wald-type confidence interval (Wilson, 1927, DasGupta et al., 2001) for a selected target confidence value $\alpha \in \,]0,1[$. For the i$^{th}$ lithology in the j$^{th}$ node, we calculate:

$$\sigma = \sqrt{2}\,\mathrm{erf}^{-1}(\alpha)\left[\frac{n_c n_f}{n_c + n_f}\right]^{1/2} \tag{10}$$

where $n_c$ is the number of correct predictions and $n_f$ the number of failed predictions. In the work we present, it

becomes:

$$\sigma_i = \sqrt{2}\,\mathrm{erf}^{-1}(\alpha)\left[\frac{\tau_i(1-\tau_i)}{\sum_{i=1}^{n_l} \tau_i}\right]^{1/2}, \tag{11}$$

where erf is the error function. More details about confidence intervals as calculated here are available can be found in (Simonoff, 2003).

In this paper, we use $\alpha = 0.95$, which approximates a 95% confidence interval where the calculated probability lies in the interval defined as $[max(\tau_i - \sigma_i, 0), min(\tau_i + \sigma_i, 1)]$. This choice of $\alpha$ is subjective. Here, we

arbitrarily chose a value that it is widely used in studies involving statistics as initially proposed by (Fisher, 1925).

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
