# Peer review of "Towards plausible lithological classification from geophysical inversion: honouring geological principles in subsurface imaging"

_Solid Earth, 2019_

## Referee Comment (RC1) · Tom Horrocks (Referee) · 12 Dec 2019

**1 General comments**

In the manuscript entitled "Towards geologically reasonable lithological classification from integrated geophysical inverse modelling: methodology and application case", Giraud et al. present a method for integrating a variety of pre-existing subsurface models. The pre-existing models that were integrated in the case study are impressively

diverse, being geophysically-derived (i.e. the inversion model and its spatial gradient), structurally-derived (i.e. the geological uncertainty matrix), or otherwise produced by a traditional workflow (i.e. the inversion start model and the most likely lithology model). The estimated lithology model, which is based on all five input models, is then refined during a 'post-regularisation' step which enforces some degree of geological plausability.

This manuscript addresses an important area of research, since it is rarely the case that a single unified lithology model is computed by joint inversion of all available information, particularly in industry due to time constraints, different datasets becoming available at different times, etc. The application of the method to refine a real pre-existing lithology model is also pleasing. As my background is primarily in machine learning, my comments below mostly address the statistical aspects of the manuscript.

**2 Specific comments**

The first step of model integration relies on a self-organising map (SOM). Ignoring the post-regularisation, the SOM is effectively (1) trained on all input models (including most likely lithology), and then (2) used to produce a new lithology model by replacing each original lithology cell with the most popular lithology within its best matching unit (BMU). The lithology population ratios within each BMU is interpreted as the probability of the cell being that lithology given all input models. The following points address this use of the SOM.

*Justifying the SOM:* Upon first reading, it was unclear to me why a clustering/partitioning algorithm (SOM) was being used for a classification problem (technically re-classifying lithology). I later realised that this approach actually allows one to view updated versions of *all* input models, which is later necessary for the geophysical consistency check. Please emphasise this earlier, perhaps where you state that

"lithologies are the quantity sought for" (Page 6, Line 15).

*Lithology probabilities:* I am unaware of any research showing that probabilities taken from a SOM in this way are well-calibrated. Note that this is a separate issue to the uncertainty of these probabilities which is instead related to the variance of the SOM's probability estimates and the samples size—the probabilities could have low 'uncertainty' but still be entirely incorrect (i.e. poorly calibrated). A new figure showing a probability calibration curve would reveal if the probabilities are reliable; the "Probability Calibration curves" page in the Scikit-learn documentation provides a good overview if the authors are unaware of this technique.

*Probability uncertainties (Appenix A):* It seems unnecessary to use a Gaussian approximation here, especially since you then have to check that each SOM cell contains enough samples. As far as I can tell, a beta distribution would be the perfect fit this problem, since it is defined over $x \in [0, 1]$ and will work regardless of sample size.

*Selecting the number of SOM units:* Generally speaking, the degree of regularisation (here the number of SOM units) chosen is that which maximises a performance metric (here accuracy) on the validation set. However, I note that your accuracy continually increases as the number of SOM units increases. The causes of this could be: (1) you have not reached the point of overfitting and need to increase SOM unit count further; (2) your training set accidentally includes your validation set; or (3) the statistical problem / the validation set is too 'easy', e.g. most samples in the validation set have near-copies in the training set. Point (3) seems most likely, and may have come about if you generated your validation set completely spatially randomly, in which case most voxels in the validation set will have spatial neighbours in the training set. In this case, a validation set containing a few strategically chosen contiguous chunks of voxels might be more suitable.

Alternatively, you may want to additionally plot the geophysical misfit (equation 9) as a function of SOM cell count. This way, you could directly select the SOM size that

results in the most integrated model that is still geophysically permissable.

The following points address other aspects of the paper.

*Geophysical consistency:* The geophysical consistency is checked by comparing the updated model's response with the original inversion model's response and checking that they differ by at most the noise threshold. However, the threshold check is an model-wide average one (based on the L2-norm), and so it's possible that the response at a few stations is significantly different (beyond the noise threshold). Can you provide some summary statistics on these (if they exist)?

*Figure 9:* What I think is missing from this figure, or perhaps the accompanying prose, is an illustration of how many voxels were 'inclusions' (where all neighbouring voxels were different lithologies), and how many voxels were changed due to the other adjacency constraints. Currently, it's impossible to know how much of an effect the interface constraints are actually having.

**3 Technical corrections**

- In-text citations for erroneous trailing commas (e.g. page 1, line 27) and spacing errors (e.g. page 2, line 14).

- When a reference is referred to as a noun, a textual citation is preferable. For example, *"In addition, apart from (Zhao et al., 2017)..."* is more correctly written as *"In addition, apart from Zhao et al. (2017)..."* or similar.

- Page 1, line 11: "apply it to field data"

- Page 2, line 28: facieses → facies

- Page 4: The reason for the $H$ subscript in $W_H$ is never explained; a short explanatory note would be nice (is it related to entropy)?

---

## Referee Comment (RC2) · Anonymous Referee #2 · 20 Dec 2019

**1  General Comments**

The authors present a new method to estimate likely subsurface lithologies and associated uncertainties by applying self-organizing maps to geophysical (a 3D density model), geological (a 3D lithology model), and derived model products (density model gradient, etc.). With the aim of quantifying uncertainty in the potential lithology, the method presented is novel and a welcome contribution to the existing literature on this topic as it relates to resource exploration. My background is geophysics and many of

my comments are in relation to those aspects, whereas aspects related to the machine learning are far afield for me.

I think the title containing 'integrated geophysical inverse modelling' is misleading, as only a single geophysical model is used in the work. Further to that, geophysical inverse modelling is not the main focus of this work that simply uses a density model obtained from geophysical inverse modelling. I suggest a title along the lines of 'Towards geologically reasonable lithological classification from geologic and geophysical models: methodology and application'.

Throughout the text, there are multiple phrases that refer to the same thing. For ease of reading, I would suggest selecting one phrase and using it throughout so there is less opportunity for confusion on what exactly is being referred to. For example, 'inversion results (Page 2, line 1)', 'inverse modelling results (Page 3, line 7), 'geophysical inverse models', 'geophysical models', and 'model' all seem to refer specifically to the density model from geophysical inversion of gravity data. Selecting a single phrase to refer to this association would be helpful.

The text was somewhat difficult to follow, and I found myself flipping through multiple pages either forward or backward as sections were referenced throughout the text. It seems that the structure of the text could be streamlined, and the complex outline could be simplified. Additionally, some section headings were not needed while others suggest the section will deliver more than it does. There are multiple instances of a section heading followed by a subsection heading, with no text or discussion introducing the reader to the flow of the text and sections to follow. I find this approach to sections and structure confusing and a deterrent to continued reading. Specifically, '2.1 Integrated geophysical modelling' is broken into two sections that could be combined into one section on 'Geophysical and geological modelling' since most of the details within the few paragraphs that compose the two sections area left as references to other works.

As I read the content within '2.2 Classification using SOM and uncertainty analysis',

[Figure]

I found that I was jumping from section to section (often ahead) for needed context. This section seems to be overly fragmented and content should be shifted to reduce the number of forward/backward section references. For example, the mean quantization error (Q) is stated as the metric used in '2.2.2.1 SOM specifications and training procedure' but the technical introduction to the metric is left to section '2.2.2.2 Training and testing datasets'. The content within these two sub-sub sections could easily be rearranged into one section under the single section of 'SOM training and validation'. Further, the paragraph under '2.2.1 Fundamentals of SOM' could readily be moved under '2.2 Classification using SOM and uncertainty analysis' (which currently is a lone section header with no text).

**2  Specific Comments**

- Paragraphs in the introduction seem piecemeal and could use some editing to link the topics together. Perhaps consider including a summary paragraph that will serve two purposes: tying all the topics included in the introduction together and providing a roadmap for what is to come in sequence within the rest of the manuscript.

- Would Figure 1 showing the method be better placed in the methods section? Placed in the introduction with a single sentence reference the figure with no discussion only causes me confusion as to what I'm supposed to learn from the diagram in the context of the introduction. The diagram itself is also a bit confusing. Is there a purpose for listing the numbers 1-5 and a-c? Data, concepts, and methods are all mixed in the diagram - might be clearer if the boxes for each were distinguishable from each other (e.g. inputs, outputs, algorithms).

- Would it be worthwhile explicitly stating that there are two separate examples, a synthetic and field case?

- Part of the method to estimate uncertainty is predicated on the fact that the post-regularisation step results in a density model that is within the null space in a geophysical sense. This is a good metric to show that changes in lithology are within the uncertainty tolerance of the individual density model being used, however this does not address the broader aspect of uncertainty where the actual density model is concerned. The density model is, as pointed out, a deterministic model. Given the non-unique nature, if the specific density model is not correct than any perturbations in the null space based on the model are also in error. How do you account for the fact that the input data (density model) may be a major source of uncertainty (it could be altogether incorrect)?

- The only geophysical data used in this method and example is gravity data and the density model resulting from geophysical inversion. This does not become clear until well into the text. Consider explicitly stating this earlier on in the text, perhaps in the methods section where the geophysical inversion is introduced.

- How are lithology changes mapped back into density changes for the forward model comparison? Is it a range for the density or a distribution?

- The phrase 'integrated inversion' is used throughout the text. It seems to me that the type of geophysical inversion used for this work is typical single data set inversion. I do not think the term 'integrated' is applicable here.

- In text citations do not need parenthesis around the citation.

- Citation punctuation throughout needs attention, as there are unnecessary commas and periods throughout.

- In the References section, many of the references are missing the journal or publisher (i.e. Lelievre and Farquharson (2016); Lindsay et al. (2018); Meju and Gallardo (2016); Moorkamp et al. (2016)).

- Throughout the text, 'self-organizing maps (SOM)' appears. Introducing the abbreviation SOM should only be necessary in the first instance of fully writing out 'self-organizing maps'.

- There are several instances where 'This' or 'That' is used to begin a sentence. While it is appropriate English, I find it difficult to follow when this is the norm rather than the exception. An example of this is on Page 4, Line 29; 'This metric' could be replaced with 'Shannon's entropy' with little increase in text while making the reading easier.

- I have found the use of 'lithological model', particularly in the results section, confusing as this could refer to the initial model used for training; the predicted model; or the model generated from geologic observations. I suggest identifying three different phrases to consistently use and refer to each of these models to eliminate any possible confusion.

**3  Technical corrections**

- Page 1

  – Line 25-30: Please expand on the 'inherent duality' of geology and geophysics
  – Line 30: What are the geophysical quantities modelled from petrophysical and geological information?
  – Line 35: citations should be in chronological order?
  – Line 35: 'which consist in' should be 'which consist of'

- Page 2

- Line 4: Both e.g. and etc. imply a subset, use one or the other
- Line 5-10: Language 'on the one hand', 'on the other hand', 'Nevertheless, like all modelling results' make it sound to me like these are opinions
- Line 15: Is it possible to provide examples of the 'broad range of parameters'?
- Line 15: 'results' should be result
- Line 26: 'informed' should be inform
- Line 28: need 'to' or 'for' in between 'consideration geological'
- Line 30: Are you 'mitigating' that fact that 'no consideration is given to geological information and rules' or providing a method to address the gap?
- Line 30: 'partially addresses the issues and shortcomings': please elaborate on the aspects still be tackled by you and the geoscience community
- Line 35: Is this a 'fully controlled environment'? Using a 'semi-synthetic' dataset (Page 3, Line 15) implies there may be some unknowns in this environment.

- Page 3

  - Line 3: The phrase 'can serve multiple objectives' I think might be a bit misleading as these are not necessarily objectives of the method; as written it sounds like the first two items stated in this paragraph are motivating goals for why you have developed the method as it is. The third is an example of the method rather than an objective of the methodology.
  - Line 12: The a or b reference for Kohonen (1982)?

- Page 4

  - Line 9: 'approach which' should be 'approach where'

– Line 12: 'd represents measurements'; 'calculating the data'; the terminology for these should be consistent. Call d 'observed data' and then use 'predicted data' or similar consistent phrasing. 'Measurements' is not a common word when referring to geophysical data and in context, I took it to mean any measurement geophysical/geological when this is actually a reference to geophysical data only.

– Line 15: How is $W_m$ specified?

– Line 25: 'geological models is' should be 'are'

– Line 29: Move citation Wellmann and Regenauer-Lieb (2012) to line 28 just after 'can be estimated'

• Page 5

– Line 1: What is meant by soft?

– Line 14: What exactly are the 2D maps?

– Line 20-22: Provide references for the Q elbow curve rather than the L-curve.

– Line 26: Will the fact that this is using semi-synthetic data have an effect? Will it alter the results?

– Line 30: Where does the starting model for inversion come from?

– Line 35: By 'datum', do you mean 'variable'? What is u(5)?

• Page 6

– Line 1: 'interpretable datasets'; What is an example of a non-interpretable dataset?

– Line 7-15: The text on the Q metric could be moved to the SOM fundamentals or where it is referred to a page earlier.

- – Line 20: remove 'are' in which are cannot'
- – Line 30-33: 'post-regularisation, which consists of'; typo 's' at the end
- – Line 34: Use of PR to reference post-regularisation could be defined the first time this phrase is used (in the introduction).

- Page 7

  - – Equation 3: What is $u_k$?
  - – Equation 4: What is $l$ in $n_l$?

- Page 8

  - – Line 19-20: "with index B with 20% accuracy", what does this mean?
  - – Line 21: 2.2.4 Estimation should be Estimating

- Page 10

  - – Figure 3: is the dashed line showing sub-basin extent black or red?

- Page 12

  - – Line 2: "the inverted model and its spatial gradients" is restated in the next sentence

- Page 14

  - – Line 7: "It" should be "In"

- Page 19

  - – Line 5-8: Include in the literature review in the introduction rather than here.

- Page 21

– Line 12: "are available" can be removed

---

## Author Comment (AC1) · 4 Feb 2020

Dear Reviewer,

Thank you for your insights and comments. We have implemented most of the recommendations you made to the exception of a few where we thought that an alternative was possible and another one where we think that a clarification accompanied with a modification of the prose suffice. Seeing that some information that were not essential to the paper had the potential to confuse the reader, we decided to remove the

incriminating bits. This removal alters neither the meaning nor the main message of the paper. Please find below our detailed answer to the different points you raised. To refer to a specific paragraph of comment, we reproduce the title you wrote in *italics*, followed by our answer.

**Detailed answer:**

*Justifying the SOM:* We have added what you suggested (page 6 line 15 on the non-revised manuscript). We added a sentence to the text of the manuscript: "In our case, the utilisation of SOM for partitioning the input models allows the recovery of lithology, which is a geological quantity reflective of all input data. It is also useful in that, as we will see later, the consistency of the recovered lithological model can be analysed from a geophysical point of view."

*Lithology probabilities:* We agree with the point raised here. We have adjusted the wording. We replaced 'lithology probability' with a more appropriate term. Instead of using 'lithology probability', which we calculated as the 'relative apparition frequency' for each unit of the SOM, we use 'prediction accuracy' of that same unit. In this case, the former and the latter are interchangeable as they are calculated in the same fashion (equation 7). We think that using prediction accuracy is clearer and more appropriate in the context of a paper relying on SOM.

For the sake of simplicity, and to maintain the accessibility of the manuscript to a broad audience, we plotted the box-plot of the prediction accuracy for the different lithologies as it complements the prediction accuracy for each lithology. We think that such statistics are a useful addition to Figure 10 that provide the reader with summary statistics and an estimate of the overall confidence one can place in the recovered lithologies volume without looking in the details. It also complements the results shown in the evolution of the prediction accuracy as a function of the number of elements in the SOM. Since this may be useful but is not essential to the manuscript, we added it to the appendices (Appendix B).

*Probability uncertainties:* The 'probability uncertainties' are not essential to the manuscript and removing it from the document does not alter the message significantly. Therefore, we have removed it from the paper. Future work in this direction would be an interesting way to infuse a little bit of real-world physics into simple machine learning algorithms like SOM; however, since this idea did not come from us we did not add it to the manuscript. Information about the positive recovery rate is provided for the reader by the boxplot mentioned above. In the revised version of the manuscript, the assessment of the prediction accuracy is shown in the appendix where the box plot of prediction accuracy is shown. We have added the following in section 3.2.3: "For completeness, assessment of the prediction accuracy of the different lithologies is shown by the corresponding boxplot in Appendix B".

*Selecting the number of SOM units:* We agree with your comments about the evolution of the prediction accuracy. However, in the case we present, the prediction accuracy increases until reaching approx. 750 units. Beyond this point, it stabilizes and oscillates around the maximum values. This is not obvious from the Figure. To lift this ambiguity, we have added the following statement to the legend of Figure 8: "Note that after 750 units, the quantization error for the different lithologies stabilizes and oscillates around its maximum values".

*Geophysical consistency:* We agree that additional information would help support the point we make about geophysical consistency and have added more information about the comparison between the updated model's response and the original inversion model's response. To compare how they differ we have added to the manuscript the misfit maps – before and after classification. We have added the corresponding maps in Appendix. Note that the data misfit maps are, overall, similar.

*Figure 9:* We have added the number of cells that were modified. The following was added to the manuscript: "a total of 2561 inclusions was identified" in the text accompanying the figure.

*Additional modifications of the manuscript:* to improve readability, we made a few minor modifications to the manuscript, dealing with wording.

---

## Author Comment (AC2) · 4 Feb 2020

Dear Reviewer,

Thank you for your comments and remarks about the manuscript. We have addressed your comments and followed most of your suggestions. We found them useful in improving the manuscript.

In what follows, we first provide a point by point answer to your general comments,

and then to the specific comments you made. Finally, we go through the technical corrections.

**General comments**

*Title.* We have changed the title of the manuscript accordingly with your comment.

*Lexicon about previous modelling.* Phrasing about 'inverse modelling results', 'geophysical inversion results' , 'inverse modelling results', 'inverse modelling results', that refer to the same thing. We have replace these terms by 'inversion results'.

*Text structure.* We have simplified the text structure following your suggestions. Sections "2.2.1. Geophysical inversion scheme" and "2.2.2 Geological uncertainty" were merged under "2.1 Geophysical and geological modelling". The sub-subsection 2.2.3 introducing post-regularisation was move up one level to become 2.3. and we added an introductory sentence directly below the title: "This subsection introduces the post-regularization scheme used in this work and details its implementation and usage in the workflow introduced here".

We moved 2.2.4 and 2.2.5 into the under "2.4 Uncertainty analysis", and adjusted the name of the subsection to make it more reflective of the contents. Having fewer sub-subsections also addresses your comment about the lack of introductory material between section and subsection headers.

We re-arrange the content introducing SOM as you suggested: the second paragraph of the corresponding section "In the approach we follow, the optimum number of neurons..." was moved to the end of the section so it follows the logical order of the process more closely.

**Specific comments**

*Paragraphs in the introduction.* We brought minor alterations to the text in several paragraphs in the introduction and reduced the number of "this" (see comments about usage of the word "this" below), which we think helps improving the readability of the
introduction. We have also added a paragraph at the end of the introduction that summarises the paper while introducing its general structure:

"The rest of this paper develops as follows. Section 2 provides the theoretical background necessary to reproduce the work presented. It first briefly describes the geophysical and geological modelling schemes (subsection 2.1) used to obtain the models that are used as input for classification using SOM (subsection 2.2). Post-regularization as applied to such classified lithologies (subsection 2.3) and the related uncertainty analysis in terms of prediction accuracy and geophysical consistency are then detailed (subsection 2.4). Following this, Section 3 presents an application case using data from the Yerrida Basin (Western Australia). Geological and geophysical modelling results are first summarized and the rules defining post-regularisation operator in the area are introduced (3.1). The classification of results from geological and geological modelling and post-regularization are then presented alongside the related uncertainty analysis, supporting a potential re-interpretation of the geological model of the area (subsection 3.2). The discussion and conclusion sections follow and complete this contribution."

*Figure 1.* We have moved Figure 1 to the methodology section as you suggested. It now appears in *subsection "2.3.2 Implementation"* after it is referred to in the text.

*Concern/question about impact of uncertainty in density model: "How do you account for the fact that the input data (density model) may be a major source of uncertainty (it could be altogether incorrect)?"* We do not address the uncertainty in the density model directly. However, in the training set, the density model we used is the one that is recovered from inversion of synthetic data using the same parameterization of the geophysical inverse problem as for the inversion of field data only. So, the uncertainty in the density model is indirectly accounted for in the training phase and the assumption is made that field data inversion results are affected in the same way. We assume that non-uniqueness and measurement uncertainty are affecting field gravity data in the same way it does affect the synthetic case, which is does to a large extent since the

assumed noise in the data are the same, are is the data acquisition setup.

We have added the following in the discussion:

"While we do not address the uncertainty in the density model directly, we assume that non-uniqueness and measurement uncertainty affect both field data and synthetic data in the same manner due to the noise component and parameterization of each being the same."

We have stated in the text that we deal with the inversion of gravity only, and gravity modelling is now mentioned at the end of the introduction.

*'How are lithology changes mapped back into density changes for the forward model comparison? Is it a range for the density or a distribution?'* The lithologies are mapped back into density by assigning the corresponding BMU's value in place of the lithology. In the case of post-regularisation (PR), the value assigned corresponds to the closest unit from the SOM that does not violate the geological rules that are enforced, hence the modification of the recovered density contras generated by PR.

*The phrase 'integrated inversion' is used throughout the text. It seems to me that the type of geophysical inversion used for this work is typical single data set inversion. I do not think the term 'integrated' is applicable here.* We have replaced the two occurrences of the term 'integrated inversion' by 'geophysical inversion' to maintain generality.

We followed the suggestions you made regarding citations and references.

*Throughout the text, 'self-organizing maps (SOM)' appears. Introducing the abbreviation SOM should only be necessary in the first instance of fully writing out 'self-organizing maps'.* The word SOM appears 30 times while string of words "self-organizing maps (SOM)" appeared only three times. The aim was to make it clear for the read that SOM = self-organizing maps. Following your comments, we kept only the first instance. Overall, the string of characters "self-organizing maps" appears a total of 15 times in the manuscript, 2 of which are in the main body of the text.

*Usage of words 'this' and 'that'.* We have reduced the number of occurrences of the word 'this' from 76 to 63. It did not seem to us that the word "that" was overused so we have not tried to reduce the number of times it was used.

*"I have found the use of 'lithological model', particularly in the results section, confusing as this could refer to the initial model used for training; the predicted model; or the model generated from geologic observations. I suggest identifying three different phrases to consistently use and refer to each of these models to eliminate any possible confusion."* To clarify things, we have added the words 'training' and 'predicted' wherever we felt that necessary to make it clearer to the reader.

**Technical corrections**

*Page 1*

*– Line 25-30: Please expand on the 'inherent duality' of geology and geophysics.*

We have modified this statement, and replaced it by: 'fundamental complementarities [. . .] in modelling the same object (the earth)'

*– Line 30: What are the geophysical quantities modelled from petrophysicaland geological information?*

*We added the following information:* "(seismic velocities, mass-density, etc.)".

*– Line 35: citations should be in chronological order?* Yes we put them in chronological order. *– Line 35: 'which consist in' should be 'which consist of'.* Done.

*Page 2*

*Line 4: Both e.g. and etc. imply a subset, use one or the other.*

Done.

*– Line 5-10: Language 'on the one hand', 'on the other hand', 'Nevertheless, like all modelling results' make it sound to me like these are opinions.* We have removed 'on

the one hand', 'on the other hand' and replaced 'nevertheless' by 'however'.

*– Line 15: Is it possible to provide examples of the 'broad range of parameters'?*

We have rephrased the incriminated sentence to make it clearer: 'It is also evident that lithologies (or facieses) are characterised by a broad range of properties that are the result of complex, non-linear physical processes'

*– Line 15: 'results' should be result.*

Done.

*– Line 26: 'informed' should be inform.*

Done.

*– Line 28: need 'to' or 'for' in between 'consideration geological'.*

Done.

*– Line 30: Are you 'mitigating' that fact that 'no consideration is given to geological information and rules' or providing a method to address the gap?*

We have rephrased the sentence as "To complement existing methodologies, we propose a solution that partially addresses the lack of consideration given to geological information during classification."

*– Line 30: 'partially addresses the issues and shortcomings': please elaborate on the aspects still be tackled by you and the geoscience community.*

To avoid a lengthy explanation we have removed this statement (see rewritten sentence above, first comment about page 1).

*– Line 35: Is this a 'fully controlled environment'? Using a 'semi-synthetic' dataset (Page 3, Line 15) implies there may be some unknowns in this environment.*

We use the term 'semi-synthetic' because the synthetic dataset is derived from a geological framework built using field geological measurements.

To make things clearer we have added the following (underlined in the original sentence):

"Using an artificial neural network trained in a fully controlled environment (all variables in the model used for training being perfectly known) with attributes characterizing the geophysical inverse model."

*Page 3*

*– Line 3: The phrase 'can serve multiple objectives' I think might be a bitmisleading as these are not necessarily objectives of the method; as writtenit sounds like the first two items stated in this paragraph are motivating goals for why you have developed the method as it is. The third is an example of the method rather than an objective of the methodology.*

Correct. We have modified the text to: 'The methodology we propose can serve two main objectives'.

*– Line 12: The a or b reference for Kohonen (1982)?*

Both actually. Done.

*Page 4*

*– Line 9: 'approach which' should be 'approach where.*

Done.

*Line 12: 'd represents measurements'; 'calculating the data'; the terminology for these should e consistent. Call d 'observed data' and then use 'predicted data' or similar consistent phrasing. 'Measurements' is not a common word when referring to geophysical data and in context, I took it to mean any measurement geophysical/geological when this is actually a reference to geophysical data only.*

Done. We have replaced 'measurements' by 'data'.

– *Line 15: How is* Wm *specified?*

We have added the following information:

"here, $W_m$ is the identity matrix".

– *Line 25: 'geological models is' should be 'are'.*

We changed 'series' to 'ensemble'. Then we keep 'geological models is'.

– *Line 29: Move citation Wellmann and Regenauer-Lieb (2012) to line 28 just after 'can be estimated.*

Done.

*Page 5*

– *Line 1: What is meant by soft?*

That was a typo, legacy from a previous version of the manuscript.

– *Line 14: What exactly are the 2D maps?*

To make the explanation clearer we have enriched the previous paragraph with the underlined information:

"This map, which can be 2D or 3D, is made of a predefined number of interconnected neurons (also referred to as 'nodes' or 'units') that have a fixed network configuration. Projection occurs during the training phase, where the locations of the neurons in the manifold are iteratively adjusted so that they approximate it optimally. "

And to make the text more readable we have deleted the section headers separating this paragraph from the next.

– *Line 20-22: Provide references for the Q elbow curve rather than the Lcurve.*

We have rephrased the explanation as: "Note that we apply the same principle as the well-known L-curve principle".

*– Line 26: Will the fact that this is using semi-synthetic data have an effect?Will it alter the results?*

Using field-data only would lead to potential bias in the classification in that some input may be affected by, for example, lithological interpretation errors. Using the semi-synthetic dataset as we do here alleviate this issue. We have adapted the text as follows:

"We utilize this controlled environment to estimate the accuracy our predictions for each class identified in the studied volume without the errors associated with well positioning or lithology interpretation errors."

*– Line 30: Where does the starting model for inversion come from?*

The following information was added: "The starting model is obtained from prior information. Here, it is the expected petrophysical property model from geological modelling."

*– Line 35: By 'datum', do you mean 'variable'?*

*We brought the following modifications to clarify this (underlined).* "Each datum from the training and tests datasets is a vector $x \,\epsilon\, \mathbb{R}^{n_v}$, with a number of variables $n_v = 5$, where $x\,(5)$"

*Page 6*

*– Line 1: 'interpretable datasets'; What is an example of a non-interpretable dataset?*

We have removed the words *'interpretable datasets'*.

*– Line 7-15: The text on the Q metric could be moved to the SOM fundamentals or where it is referred to a page earlier.*

*Line 20: remove 'are' in which are cannot'*

Done. *– Line 30-33: 'post-regularisation, which consists of'; typo 's' at the end*

*– Line 34: Use of PR to reference post-regularisation could be defined the firsttime this phrase is used (in the introduction).*

We do not use 'PR' anymore and just use the full term 'post-regularization'.

*Page 7*

*– Equation 3: What is* uk*?*

*– Equation 4: What is* l *in* nl*?*

We have clarified the equation.

*Page 8*

*– Line 19-20: "with index B with 20% accuracy", what does this mean?*

*We have rephrased the sentence as: "*For instance, in a two-lithology scenario, a given node may be found to predict lithology A using the validation dataset correctly 80% of the time (80% accuracy) and lithology B with correctly 20% of the time (20% accuracy)."

*– Line 21: 2.2.4 Estimation should be Estimating*

We changed the title for 'Prediction accuracy of the recovered lithologies'

*Page 10*

*– Figure 3: is the dashed line showing sub-basin extent black or red?*

Rewritten: "The red dashed line outline the modelled area"

*Page 12*

*– Line 2: "the inverted model and its spatial gradients" is restated in the next sentence*

Addressed.

*Page 14*

**–** *Line 7: "It" should be "In"*

Done.

*Page 19*

**–** *Line 5-8: Include in the literature review in the introduction rather than here.*

*Page 21*

*Line 12: "are available" can be removed.*

The section was removed (see answer to the other reviewer).

---

## Author Response (AR2)

Dear Editor,

We agree with your suggestion to modify the title, and adapted it following you recommendation. The title of the manuscript has been updated:

"Towards plausible lithological classification from geophysical inversion: honouring geological principles in subsurface imaging"

Thanks, and best regards,

The authors